

# A distributed algorithm for solving large-scale p-median problems using expectation maximization

Harsha Gwalani[1], Joseph Helsing[2], Sultanah M. Alshammari[3], Chetan Tiwari[4] and Armin R. Mikler[5]

[1] Department of Computer Science and Engineering, University of North Texas, Denton, Texas, United States
[2] Department of Electrical and Computer Engineering, Stevens Institute of Technology, Hoboken, New Jersey, United States
[3] Center of Research Excellence in Artificial Intelligence and Data Science, King Abdul Aziz University, Jeddah, Saudi Arabia
[4] Department of Computer Science and Department of Geosciences, Georgia State University, Atlanta, Georgia, United States
[5] Department of Computer Science, Georgia State University, Atlanta, Georgia, United States

## ABSTRACT

The p-median problem selects $p$ source locations to serve $n$ destinations such that the average distance between the destinations and corresponding sources is minimized. It is a well-studied NP-hard combinatorial optimization problem with many existing heuristic solutions, however, existing algorithms are not scalable for large-scale problems. The fast interchange (FI) heuristic which yields results close to the optimal solution with respect to the objective function value becomes suboptimal with respect to time requirements for large-scale problems. We present a novel distributed divide and conquer algorithm, EM-FI, to solve large-scale p-median problems quickly even with limited computing resources. The algorithm identifies the existing spatial clusters of the destination locations using expectation maximization (EM) and solves them as independent p-median problems using integer programming or FI concurrently. The proposed algorithm showed an order of magnitude improvement in time without the loss of quality in terms of the objective function value on synthetic and real datasets.

## INTRODUCTION

The p-median problem is the most commonly used formulation for solving distance-based location models. Distance-based location models are characterized by minimization of costs while locating facilities (or sources) to satisfy customer demands. In location science, costs are usually associated with travel time between sources/facilities and respective destinations/demand points. Therefore, minimization of the distance between the demand points and corresponding facilities is the goal in distance-based location models. The p-median problem specifically seeks to locate $p$ facilities and allocate destinations to them

Corresponding author
Harsha Gwalani,
harsha.gwalani@gmail.com

such that the average distance between the facilities and destinations is minimized, and each of the total $n$ destinations is assigned to a facility. This NP-hard problem (*Kariv & Hakimi, 1979*) has been used to solve various industrial applications such as location and demand distribution of warehouses (*Baumol & Wolfe, 1958*; *Dejax, 1988*), shopping centers (*Nwogugu, 2006*), fire stations (*Plane & Hendrick, 1977*), *etc*. Portions of this text were previously published as part of a preprint in *Gwalani et al. (2022)*.

Many heuristic and meta-heuristic algorithms have been proposed in the literature to solve this problem approximately. However, these approaches do not scale well as the number of destinations and the number of sources to be selected increase. Evaluation and comparison of existing heuristic algorithms in *Gwalani, Tiwari & Mikler (2021)* showed that the execution time required to achieve close to optimal solutions for a problem with $n > 3,000$ and $p > 500$ exceeds 3 h on a reasonably equipped personal computer. Additionally, the execution time increases with increases in both $n$ and $p$. This problem can be solved exactly using a mixed integer linear programming (MILP) solver, however, this approach becomes infeasible with respect to both execution time and memory requirements as the scale of the problem increases. Very few studies exist in the literature that solve truly large-scale p-median problems. Table 1 [1] lists the scale of the largest problems that have been solved in the literature along with the methods used to solve them. As the scale of these problems is still quite low for solving problems with $n > 3,000$ and $p > 500$, there is scope for research in this area.

In this article, we present a distributable decomposition-based heuristic algorithm that utilizes the spatial distribution of the destinations in the problem to allow for concurrent and faster execution. The problem is divided into subproblems using expectation-maximization clustering, and the subproblems are solved concurrently using MILP or the fast-interchange (FI) heuristic algorithm. The subproblem solutions are then combined by reassigning destinations to their closest facility across all subproblems. This approach results in significant time improvements over the existing serial algorithms without the loss of the quality of the solution. The quality of a solution is measured by the demand weighted average distance between the destinations and corresponding facilities. We evaluate the proposed novel algorithm on synthetic as well as real datasets. Location of *ad-hoc* clinics in a region to dispense pill bottles, vaccines or other medical resources to the affected population in case of a bio-emergency is used as the application for real datasets. The algorithm is used to locate these clinics and map them to individuals (or population centers) so that all the affected individuals can receive the resources in a timely manner.

Our analysis of optimal or close to optimal solutions for p-median problems showed that there is a relationship between the selected facilities and the spatial clusters of demand points in the problem space. A cluster is a subregion in the region with a high density of demand points. It was observed that a demand point belonging to a cluster is not served by a facility located in another cluster, if $p$, the number of facilities to be selected, is greater than the number of distinct clusters in the region. This is true in theory, since the centroid of the demand points in a cluster is chosen as the facility location, then any demand point will always be closer to its own cluster centroid than a different cluster centroid, the facility located in another cluster. However, in real world scenarios, exceptions to this observation

---

[1] We have tried to include results from all well-cited articles in this area. The table is not meant to compare computation time because of variables such as computing resources and implementation differences.

**Table 1 Scale of p-median problems solved using heuristic or meta-heuristic solutions existing in the literature.** $n$ is the number of demand points in the region and $p$ is the number of facilities to be selected.

| Problem scale ($n, p$) | Method & Year | Time (in seconds, rounded to closest integer) |
|---|---|---|
| (500, 30) | Global Regional Interchange Algorithm (GRIA) (*Densham & Rushton, 1992*) | 865 |
| (500, 20) | Tabu Search, *Rolland, Schilling & Current (1997)* | 447 |
| (3,038, 500) | Variable Neighborhood Search (VNS) (*Hansen & Mladenović, 1997*) | 21,357 |
| (300, 35) | Heuristic concentration (*Rosing, ReVelle & Schilling, 1999*) | 102 |
| (900, 90) | Simulated Annealing, *Chiyoshi & Galvão (2000)* | 628 |
| (900, 300) | Lagrangian/Surrogate Relaxation, *Senne & Lorena (2000)* | 3,212 |
| (1,000, 333) | Genetic Algorithm, *Osman, Erhan & Zvi (2003)* | 444 |
| (5,934, 1,500) | GRASP and path-relinking, *Resende & Werneck (2004)* | 2,230 |
| (900, 300) | Column generation (Lagrangian Relaxation), *Senne, Lorena & Pereira (2005)* | 1,391 |
| (89,600, 64) | Aggregation heuristic, *Avella et al. (2012)* | 5,779 |
| (900, 90) | Lagrangian relaxation, *Daskin & Maass (2015)* | 33 |
| (900, 90) | GPU Genetic algorithm, *AlBdaiwi & AboElFotoh (2017)* | 1,407 |

are possible, if the only viable facility location is located on the edge of the cluster then it is possible that a demand point from a different cluster may be closer to this facility location and hence served by a facility in another cluster. Figure 1 shows this property for a range of values of $p$, starting from $p = 2$ to $p = 128$ for 480 demand points. The boundaries in the figure represent the service areas for facilities. All demand points within the service area of a facility are assigned to that facility. This is an interesting property of the p-median problem, which has been used in the past for cluster analysis.

Clustering or cluster analysis, in general, is the process of grouping a set of objects in such a way that the objects belonging to the same group are more similar to each other than the objects belonging to other groups. These groups are referred to as clusters in the data. If these objects represent points in space, and the similarity between them is measured in terms of distance, then the clustering problem transforms into a non-demand-weighted p-median problem. In fact, the solution to a p-median problem closely resembles the solution to a clustering problem on the same data if the selected facilities are treated as cluster medians. The popular k-means clustering algorithm (also known as Lloyd's algorithm), based on the partitioning method proposed in *Lloyd (1982)* is strikingly similar to Maranzana's alternate selection and location algorithm for solving the p-median problem (*Maranzana, 1964*). Maranzana's algorithm was proposed 20 years before the k-means clustering algorithm. Moreover, researchers have used p-median approaches to discover population clusters in a region. An interchange based algorithm is used to cluster spatial entities, along with Maranzana's alternate selection and allocation in *Taillard (2003)*. *Mulvey & Crowder (1979)* use Lagrangian relaxation to improve the quality of clusters in homogeneous data and compare results with other clustering methods.

The key observation from this existing research is that p-median solutions are good clustering solutions for distance-based spatial clustering problems. We utilize the inverse of this observation to solve large-scale p-median problems. The lack of overlap between
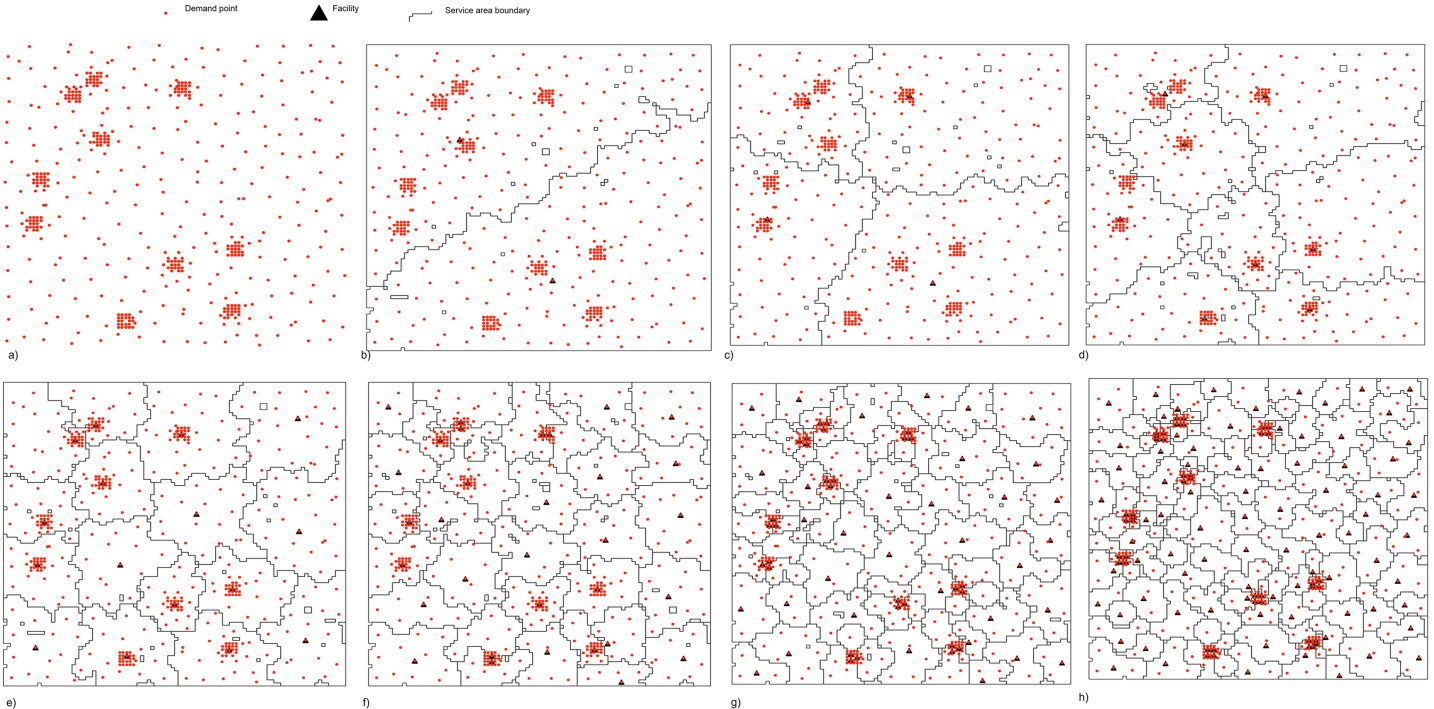

**Figure 1 Correlation between clustering and p-median solutions.** (A) Demand point locations, solutions for the p-median problem: (B) $p = 2$, (C) $p = 4$, (D) $p = 8$, (E) $p = 16$, (F) $p = 32$, (G) $p = 64$ and (H) $p = 128$. The black triangle represents a facility and the orange points represent the demand points.

service areas in one cluster with service areas of another highlights the opportunity for the use of decomposition methods for solving large-scale problems. These problems can be solved efficiently by identifying these clustered regions of no overlap and selecting facilities for them independently. Existing decomposition based techniques (*Taillard, 2003*) break the problem into arbitrary number of subproblems using various p-median heuristic algorithms. Solutions obtained from p-median algorithms are good clustering solutions only if the goal of clustering is to group data points based on distance, and not necessarily to identify dense sub-regions in the region. They are not optimal for identifying demand clusters in the region, particularly if clusters are located close to each other. This is illustrated in Fig. 2. The 10 service areas in the figures do not correspond to the 10 clusters in the region, and hence do not represent the best subproblem candidates. In this research, we extend the idea of decomposition of large p-median problems, the decomposition in the presented algorithm however, is based on the spatial configuration of demand points, therefore, the number of subproblems is governed by the structure of the problem.

We discuss the formulation of the p-median problem and some common heuristic approaches for the problem in the next section. The methodology is described in detail in "Methodology". We show results for synthetic and real datasets in "Results". Finally, we conclude the article in "Discussion".

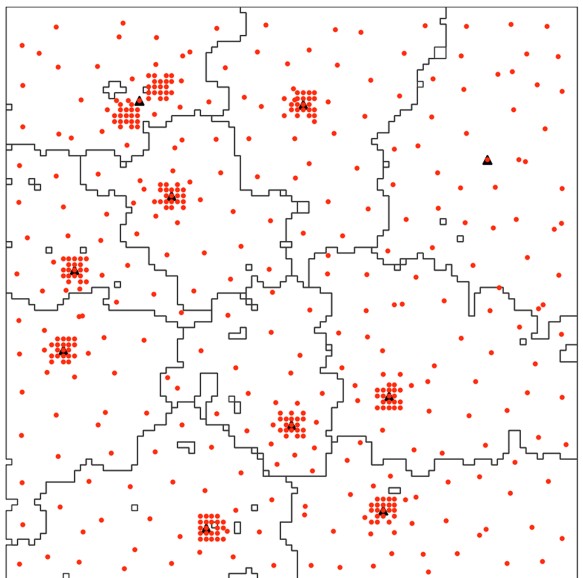

**Figure 2 Service areas corresponding to the optimal solution for $p = 10$ for demand point distribution with 10 clusters.**               

# BACKGROUND AND LITERATURE REVIEW

## p-median problem formulation

Linear integer programming (LIP) can be used to obtain optimal solutions for small scale p-median problems but these solutions quickly become time and memory prohibitive as the values of $n$ and $p$ increase. The LIP formulation (*Daskin, 2013*; *Rosing, Revelle & Rosing-Vogelaar, 1979*; *Senne, Lorena & Pereira, 2005*; *Resende & Werneck, 2004*) for the discrete p-median problem is described below.

## LIP formulation

Decision variables:

1. $X = \{X_j\}, \forall j \in \{1, 2, 3 \ldots n\}$
2. $Y = \{Y_{ij}\}, \forall i \in \{1, 2, 3 \ldots n\}$ and $\forall j \in \{1, 2, 3 \ldots n\}$

Minimize $Z = \Sigma_{i=1}^{n} \Sigma_{j=1}^{n} w_i Y_{ij} d_{ij}$ such that:
Constraints:

1. $\Sigma_{j=1}^{n} Y_{ij} = 1, \forall i \in \{1, 2, 3 \ldots n\}$
2. $\Sigma_{j=1}^{n} X_j = p, \forall j \in \{1, 2, 3 \ldots n\}$
3. $Y_{ij} - X_j \leq 0, \forall i \in \{1, 2, 3 \ldots n\}, \forall j \in \{1, 2, 3 \ldots n\}$
4. $X_j = \{0, 1\}, \forall j \in \{1, 2, 3 \ldots n\}$,
5. $Y_{ij} = \{0, 1\}, \forall i \{1, 2, 3 \ldots n\}$ and $\forall j \in \{1, 2, 3 \ldots n\}$

$X_j$ is a binary variable that, if set, represents the selection of destination $j$ as one of the facilities. $Y_{ij}$, also a binary variable, if set, represents the assignment of destination $i$ to facility $j$. $Z$, the objective function is the demand ($w_i$) weighted sum of distances between

facility $j$ and destination $i$. This distance is denoted by $d_{ij}$. The distance value $d_{ij}$ contributes to the sum only if $Y_{ij}$ is set. Constraint 1 ensures that each destination is assigned only to a single facility. The selection of exactly $p$ facilities is ensured by Constraint 2. The assignment variable $Y_{ij}$ cannot be set if a facility is not located at destination $j$, this condition is enforced by Constraint 3. This integer programming problem can be solved by using branch and cut algorithms on the continuous linear programming solution. The size of the search space for a p-median problem solution is equal to the Stirling number of the second kind (*Resource, 2020*) or the number of ways to partition a set of $n$ items into $p$ parts. The branch and bound tree in the LIP solution may degenerate into an exhaustive search leading to exponential time complexity in the worst case scenario (*Papadimitriou & Steiglitz, 1982*).

## Literature review

Since the linear integer programming solution is not scalable, many heuristic algorithms have been proposed to yield approximate solutions. Interchange-based algorithms have been shown to yield the best results out of all heuristic algorithms. *Teitz & Bart (1968)* proposed the original exchange or interchange algorithm in 1968. Modifications to the original algorithm (FI) to reduce the number of exchanges were proposed in *Goodchild & Noronha (1983)*, *Whitaker (1983)*, and *Hansen & Mladenović (1997)*. Facilities currently in the solution are interchanged with facilities not in the solution one at a time, and the solution after this interchange is evaluated. The interchange that results in the maximum decrease in the cost function is selected in each iteration. The interchange algorithm has been shown to produce close to optimal results, but even the FI algorithm is not feasible for larger datasets because of the large number of pairwise exchanges. The total number of swaps for this algorithm is equal to $(p)(n - p)$ per iteration, where $n$ is the total number of destinations and $p$ is the required number of facilities. For each swap, the facility being introduced is tested as a candidate for each destination, therefore each iteration is $O(pn^2)$. Overall complexity of the algorithm is $O(tpn^2)$ where $t$ is the number of iterations required for convergence/termination. The value of $t$ depends on the initial solution. Other common heuristic algorithms include the Global/Regional Interchange Algorithm (GRIA) (*Densham & Rushton, 1992*), the greedy addition algorithm (*Kuehn & Hamburger, 1963*) and the alternate selection and allocation algorithm (*Maranzana, 1964*). These algorithms are discussed and evaluated in detail in *Gwalani, Tiwari & Mikler (2021)*.

Several meta-heuristic algorithms have also been proposed in the literature to improve the cost function values over those obtained from the interchange based algorithms. The most common meta-heuristics like genetic algorithms (*Osman, Erhan & Zvi, 2003*), tabu search (*Rolland, Schilling & Current, 1997*; *Loranca, Velázquez & Analco, 2015*) and simulated annealing (*Chiyoshi & Galvão, 2000*; *Murray & Church, 1996*) have been formulated and implemented for the p-median problem. Most of these meta-heuristic algorithms focus on improving the solutions obtained from local search or exchange algorithms in terms of the cost function value. The gains over the cost function value when compared to the classic exchange heuristics, however, are not substantial. The time

required to compute the solutions on the other hand, is much higher. These methods do not scale well with respect to time as the numbers of destinations and facilities increase.

Taillard presents two decomposition methods to reduce run time, local optimization (LOPT) and decomposition (DEC) in *Taillard (2003)*. The LOPT algorithm selects $p$ destinations as facilities randomly as the current solution. These facilities are added to a candidate list for subproblem creation. A subproblem is created in each iteration by selecting a facility $i$ at random, and its $r − 1$ closest facilities from the candidate list. These $r$ facilities and demand points assigned to them form a subproblem that is solved using an interchange algorithm. The facilities selected from solving the subproblem are added to the candidate list if the solution (for the subproblem) has improved, else the original facility $i$ is removed from the candidate list. The algorithm terminates when the candidate list is empty. The DEC algorithm breaks the original problem into $t$ subproblems by selecting $t$ facilities in the original problem using an interchange algorithm. The service areas created for each of these $t$ facilities serve as the regions for the subproblems. The number of facilities, to be selected in each subproblem is equal to $int(p/t)$. The value of $t$ or the number of subproblems is set to $\lfloor \sqrt{p} \rfloor$ in this algorithm. It is shown in the study that the LOPT algorithm works better than DEC, however, it is sensitive to the initial solution and requires a considerable amount of time for termination. The results for large-scale problems ($n = 85{,}900$) in the study are shown relative to the best solution obtained from multiple runs of DEC, LOPT, and the ALT procedure. The ALT procedure is a variant of Maranzana's alternate selection and allocation algorithm. The performance of these algorithms for large-scale problems was not compared with algorithms that have been known to produce close to optimal results. The decomposition into subproblems in the LOPT algorithm does not take into account the spatial distribution of the demand points. Therefore, the solutions to the subproblem may not solve the original problem optimally particularly for smaller values of $r$, and it becomes time prohibitive for larger values of $r$. The LOPT algorithm was shown to produce near-optimal solutions for small scale problems in their study. These decomposition-based techniques have the following limitations: (1) the number of subproblems is calculated using a formula that does not take into account the spatial configuration of the region, (2) the number of facilities to be selected in each subproblem is not demand weighted, and (3) the subproblems are generated by solving a p-median problem which may not successfully identify clusters in the region. The decomposition methods described in this study do not utilize the inherent clustering in the region, hence they can be improved upon both with respect to the cost function value and execution time.

## METHODOLOGY

We propose to decompose large-scale p-median problems into subproblems based on the spatial distribution of the demand points in the region. The inherent demand point clusters (denser regions) present in the region serve as good candidates for the independent subproblems. These clusters can be identified by detecting changes in the spatial distribution of demand points scattered in the region.

**Figure 3 Solving a large-scale p-median problem by decomposition: flow chart.**

The steps involved in the decomposition of $(n, p)$ problem into subproblems and the combination of the subproblem solutions to create a solution for the original problem are described in the following sections. The density-based decomposition process is explained in "Density-based Decomposition". All subregions created by density-based decomposition may not represent clustered or heterogeneous regions of demand points, so we merge the homogeneous/sparse (non-clustered) subregions into the denser subregions to reduce the subproblems to clustered subregions only in a merge step ("Density-Based Decomposition: Merge Step"). The divide-conquer step is executed after the merge step to obtain a solution to the original problem ("Divide and Conquer"). The flowchart for the entire algorithm is described in Fig. 3.

## Density-based decomposition

Clusters in a region can be identified by detecting changes in the spatial distribution of demand (population) in the region. The central place theory dictates that consumer preferences in any geographic area will lead to centers of various sizes to emerge in a region (*Daniels, 2017*). Therefore, any region is likely to have population centers with different population densities. Behavioral patterns also often lead to formation of clusters in a spatial region (*Rushton, 1969*). The goal of the density based decomposition is to identify these centers and assign the demand points to its corresponding center. We model any region as being composed of different sub-regions with corresponding mean and standard deviation of population points distributed normally around the mean and standard deviation. This Gaussian mixture model (GMM) captures the inter and intra subregion density variation adequately as it is simpler and faster than other density based clustering algorithms while performing well in identifying the dense regions. The expectation-maximization (EM) clustering algorithm (*Dempster, Laird & Rubin, 1977*) has been known to yield good results for detecting Gaussian clusters with different parameters in a dataset. It is an iterative method that attempts to predict parameters for statistical models in two steps: (1) expectation, which calculates the log-likelihood corresponding to the current estimates for the parameters, and (2) maximization, in which the model changes the parameters to increase the log-likelihood value. Since the number of clusters is an input to the algorithm, EM can be tried to fit models with a varying number of clusters and each fit can then be evaluated using the Bayesian information criterion (BIC). BIC can be used to identify the minimum descriptor length that can describe the data (*Wikipedia.org, 2019*). It penalizes the addition of new parameters to the model to avoid overfitting. We use expectation-maximization to fit the data in a GMM with $c$ components, starting from $c = 2$ to $c = c_{max}$. Each demand point, $d_i$, represents a vector of size two, $\vec{x_i} = [x_{di}, y_{di}]$. The objective of the Expectation-Maximization algorithm is to obtain a GMM for the data

that represents the normally distributed sub-populations in the overall population (*Brilliant.org, 2019*). The population refers to the set of all demand points in the region, while each sub-population is a set of demand points that belong to the same bivariate normal distribution according to the model. Each component of the GMM has a mean vector, $\vec{\mu_i}$ and a covariance matrix $\Sigma_i$. The probability distribution for the model is given by *Brilliant.org (2019)*:

$$p(\vec{x}) = \Sigma_{i=1}^c \theta_i \mathcal{N}(\vec{x}|\vec{\mu_i}, \Sigma_i) \tag{1}$$

where $\mathcal{N}(\vec{x}|\vec{\mu_i}, \Sigma_i)$ represents a bi-variate Gaussian distribution (Eq. (2)) corresponding to the component $i$ with parameters, $(\vec{\mu_i}, \Sigma_i)$. The sum of the component weights corresponding to each component, $\theta_i$ is equal to one, $\Sigma_{i=1}^c \theta_i = 1$.

$$\mathcal{N}(\vec{x}|\vec{\mu_i}, \Sigma_i) = \frac{1}{\sqrt{(2\pi)^c|\Sigma_i|}} e^{-\frac{1}{2}(\vec{x}-\vec{\mu_i})^T \Sigma_i^{-1}(\vec{x}-\vec{\mu_i})} \tag{2}$$

Given a starting value for parameters, $(\theta_i, \mu_i, \Sigma_i)$ for each component in the model, the expectation step calculates the probabilities/expectation that a data point belongs to that component. The maximization step updates the parameters to maximize the overall expectation or likelihood value. Equation (3) shows the calculation of BIC for a model (*Wikipedia.org, 2019*; *Neath & Cavanaugh, 2012*).

$$BIC = ln(n)c_m - 2ln((\hat{L})) \tag{3}$$

$(\hat{L})$ is the maximized value of the likelihood function after the EM algorithm converges, $n$ is the number of data points, $c_m$ is the total number of model parameters. To detect the high demand point density sub-regions in the area, each demand point coordinate is input to the EM algorithm. The EM algorithm is executed for $c = 2$ to $c = c_{max}$, and the model corresponding to each $M(c)$ is evaluated using BIC. It was observed that BIC analysis becomes unreliable for higher values of $c$. The gain in the maximum likelihood optimization function is higher than the penalty corresponding to the increase in the number of parameters. This results in decomposition of regions into independent components that maybe served by a common facility which is detrimental to the p-median objective function. To avoid this over decomposition, the $c_{max}$ value was set at 20 in these experiments.

Models with lower BIC values are better than models with higher BIC values, hence the value of $c$ for which the minimum value of BIC is reported is selected as the number of clusters, $k$, for the region. The region is decomposed into $k$ subregions or components using the EM algorithm. Each subregion has a different $\vec{\mu}$ and $\vec{\Sigma}$ corresponding to the positions of demand points. Each of these subregions, however, may not represent a dense or clustered region as the EM detects changes in distribution. These distributions could be homogeneous or non-homogeneous. A homogeneous distribution represents a random scatter of demand points in the region. A heterogeneous distribution is a significant deviation from a homogeneous distribution of points in the region. Figure 4 shows an example of the creation of subproblems using the EM algorithm. The initial demand

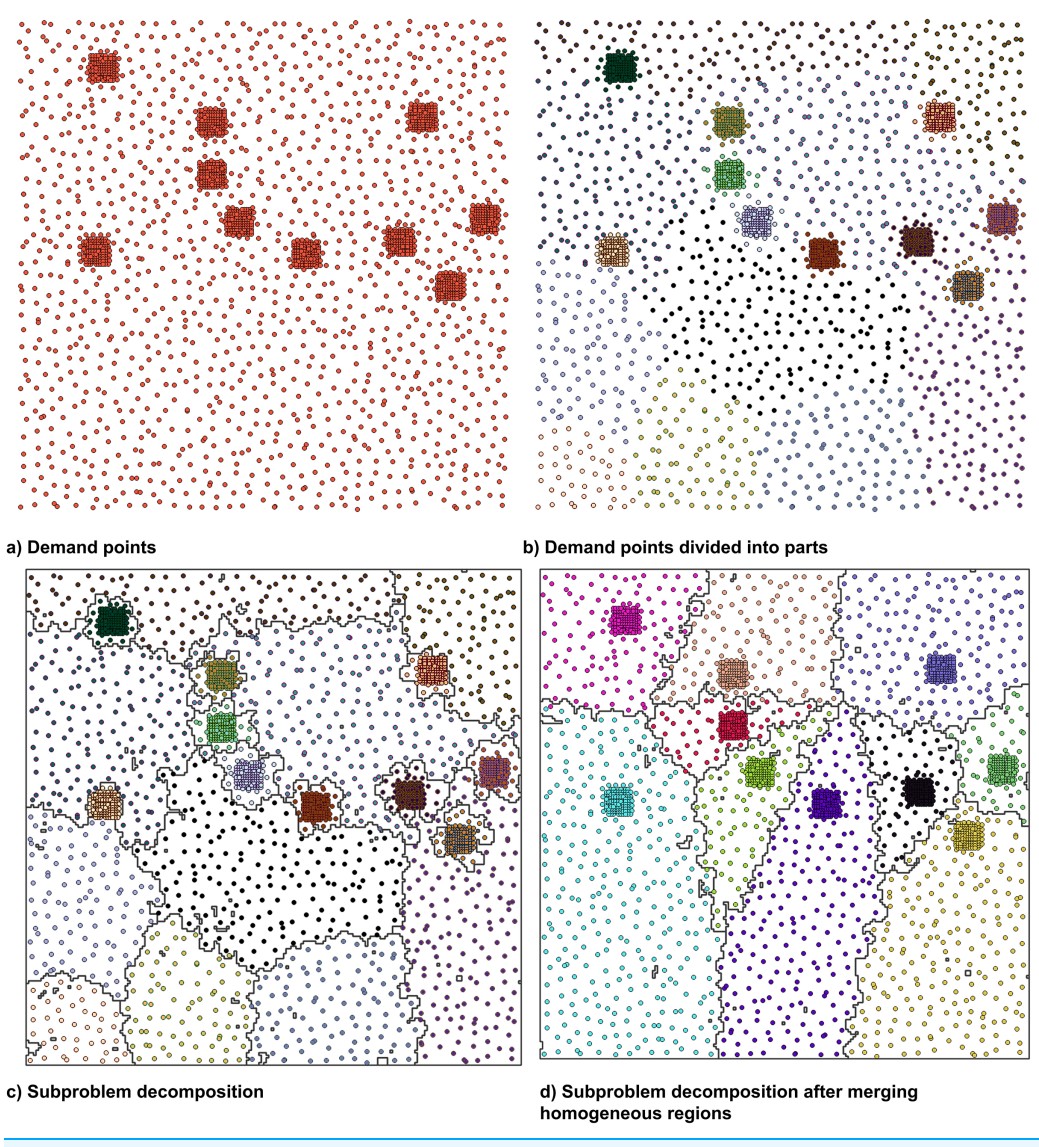

a) Demand points

b) Demand points divided into parts

c) Subproblem decomposition

d) Subproblem decomposition after merging homogeneous regions

**Figure 4** (A–D) The decomposition of a region into subregions and corresponding subproblems using Expectation-Maximization based on a Bayesian information criteria analysis.

points, membership of demand points to their corresponding Gaussian component, and the corresponding subregions are shown in Figs. 4A–4C respectively. The destinations are distributed in 20 components based on the BIC analysis. The subregions are created by unionizing the geometries corresponding to the demand points.

## Density-based decomposition: merge step

It can be seen in Fig. 4 that 10 out the 20 subregions do not represent clustered or dense areas in terms of the demand point distribution, and therefore may not be good candidates for an independent subproblem. We propose a merge step to merge the sparser and/or homogeneous regions into denser ones, thereby preventing them from being solved independently.

A subregion is labeled sparse if the demand density in the subregion is less than the demand density of the entire region. Demand density is defined as the total demand per unit area. The total demand in a subregion is calculated by aggregating the demand at all of its demand points. The heterogeneity (or homogeneity) of the spatial distribution of points in a region can be detected by comparing the existing distribution of the subregion with the expected binomial distribution using a $\chi^2$ test. The details of heterogeneity detection are explained in the appendix "Detecting Heterogeneity in a Region". Once the non-clustered (homogeneous or sparse) subregions have been identified, the demand points corresponding to these subregions are reassigned to the remaining dense subregions. Each demand point in a non-clustered subregion is assigned to its closest dense subregion. The distance to a dense subregion is measured as the distance between the demand point to be reassigned and the centroid of the subregion. Thus, the non-clustered regions are dissolved in this way reducing the number of subregions and subproblems from $k$ to $q$. Figure 4D shows the subregions corresponding to Fig. 4C after the demand points corresponding to the non-clustered subregions are merged into the dense subregions. We propose to use these subregions as independent subproblems.

If the distribution of demand points in the original problem is completely homogeneous, the merge step will lead to a single subproblem corresponding to the original problem, hence the decomposition will not succeed. This limitation of the algorithm is discussed more in "Challenges and Distance-based Decomposition".

### Divide and conquer

The distribution or density-based decomposition of the region yields $q$ subregions after the merge step is employed. The number of facilities to be selected in these subregions is determined based on the proportion of the population in the subregion with respect to the entire population. The facilities are assigned to each subregion proportional to the total demand in the subregion (integer division). The remaining facilities are assigned to the subregions in a round-robin fashion in decreasing order of demand at subregions. These subproblems are then solved exactly using MILP[2] solver if possible. If the scale of the subproblem is too large to solve using MILP, then the subproblem is solved approximately using FI. Scale based memory profiling was conducted to determine the threshold values of $n$ and $p$ to employ FI instead of MILP. The subproblem solutions are combined to obtain a global solution in the reduce step. The union of facilities selected in each subproblem forms the current solution, $X$, for the original problem. The $n$ demand points in the original problem are assigned to their closest facility in the current solution. This solution then can be further improved by applying an exchange based search algorithm (FI/GRIA) to this solution. Since the solution is near the optima, the search takes much less time than when used on a randomly chosen initial solution.

### Time complexity

The time complexity for a serial version of FI is $O(p^2 n^2)$, assuming $t$, the number of iterations, $t = cp$, for some constant $c$, where $c < 1$. This assumption is based on experimental analysis on the change in number of iterations with change in initial solution

[2] Python MILP (Mixed-Integer Linear Programming) Tools: https://pypi.org/project/mip/.

in *Gwalani, Tiwari & Mikler (2021)*. If the problem is decomposed into $q$ parts, then on average each subproblem is of scale $(n/q, p/q)$, hence the complexity reduces to $p^2 n^2 / q^4$ for all problems which are solved concurrently. The expectation maximization step is $O(n(2 + 3 + \ldots c_{max})) = O(n)$, for $c_{max} << n$ (the number of iterations needed for convergence are much smaller than $n$, hence that factor is neglected). The reassignment step is another $O(n)$ operation. The improvement step at the end is again $O(tpn^2)$ but the number of iterations required for convergence is much smaller, hence the improvement term reduces to $O(pn^2)$. We show in the article, that a reasonably good solution can be obtained even without the improvement step by using distributed computing.

## RESULTS

We compare the EM-FI algorithm to the serial implementation of the FI algorithm with respect to the cost function value and execution time for each run for synthetic and real problem sets. We evaluate two versions of the EM-FI algorithm. In "EM-FI:Reassignment", the demand points are reassigned to the facilities selected in the subproblems in the conquer step, while in "EM-FI:Improvement", the solution is further improved by using FI globally with these facilities as the initial solution. The cost function values are shown relative to the best-known solution for the problem. The best-known cost function value was calculated as the minimum of values computed using a MILP solver (time limit = 100 h), and cost values corresponding to three heuristic approaches across 30 executions. The execution time is the CPU time required to obtain the solution in each run. The execution time includes the time required to perform the density-based decomposition (with the merge step) for the EM-FI algorithm. The experiments were performed on an Intel Core i7 CPU @ 3.60 GB $\times$8 (32 GB RAM, 64-bit) machine for proof of concept, and the program allowed for solving four FI subproblems concurrently and two MILP subproblems concurrently on this machine. The numbers of subproblems should be set based on the computing resources at hand. The code for EM-FI, FI, and MILP was implemented in python. We used the python libraries, sklearn (https://scikit-learn.org/stable/modules/clustering.html) for Gaussian Mixture Models and BIC Analysis, and Scipy (https://docs.scipy.org) for chi squared tests. The FI algorithm was implemented based on the pseudo code in *Hansen & Mladenović (1997)*. The source code for existing heuristics for p-median problem along with this distributed algorithm is being made available *via* a public git repository.

### Synthetic datasets

We compare the performance of the discussed algorithm on synthetic datasets with varying scale and spatial distributions. In the synthetic datasets, the number of demand points is varied by changing the total demand in the region. All demand units are scattered in the region grid depending on the required spatial distribution. These demand units are then combined to create destinations. Each destination is assumed to have a demand selected from a normal distribution, $\mathcal{N}(1{,}500, 400)$. These values were chosen to replicate the creation of census block groups, a census geographic unit in USA. The centroid of the merged demand units for a destination defines the coordinates of the demand point. Because of the stochastic nature of the this process, not all points on the fine grid get
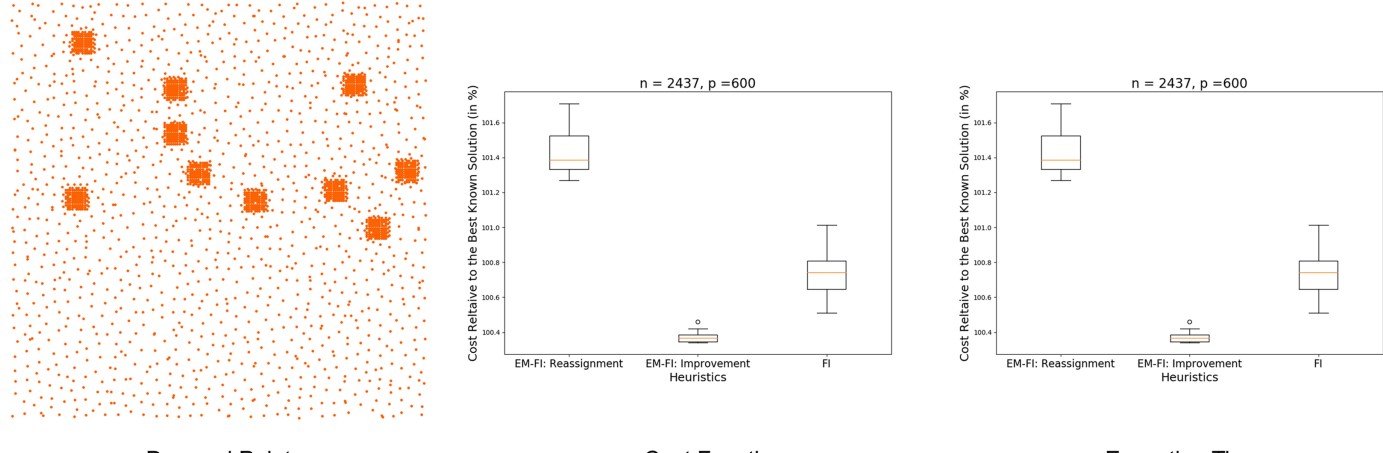

a) Synthetic Dataset I: n = 2,437, p = 600, number of clusters = 10

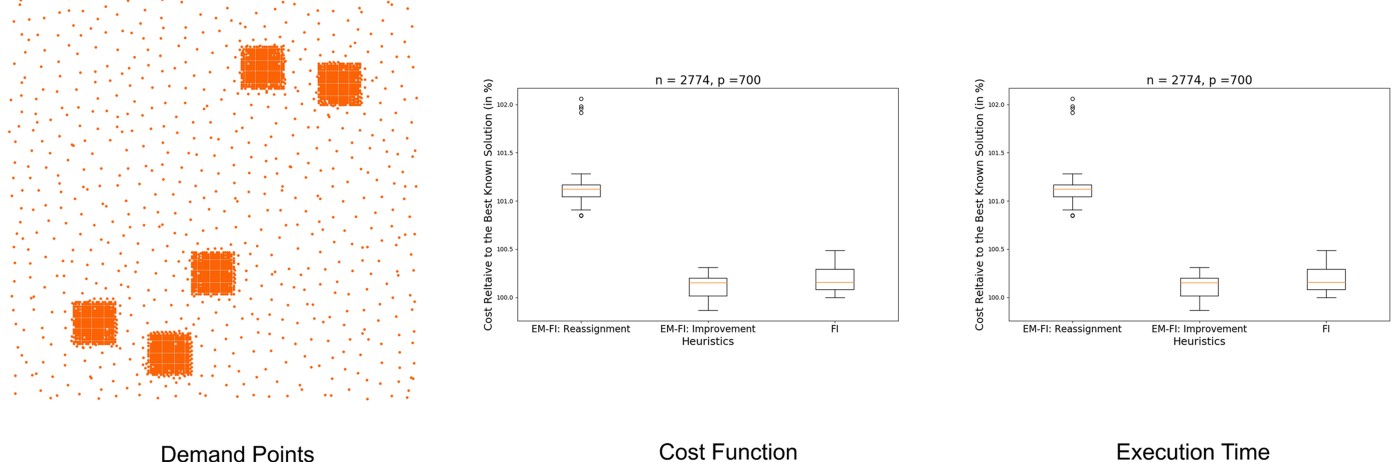

b) Synthetic Dataset II: n = 2,774, p =700, number of clusters = 5

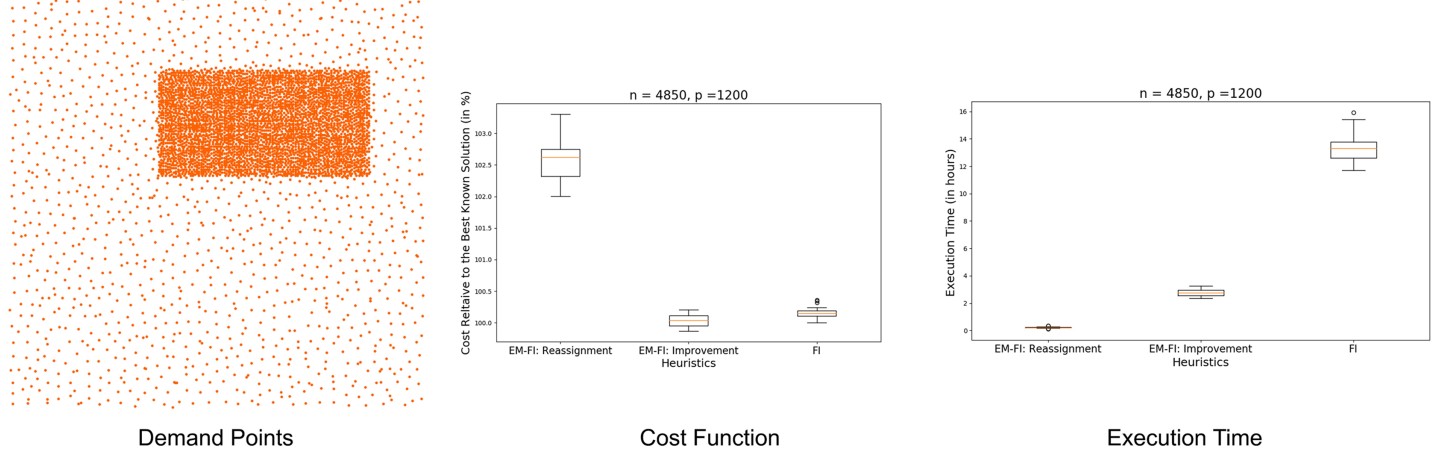

c) Synthetic Dataset III: n = 4,850, p = 1,200, number of clusters = 1

**Figure 5** **(A–C) Synthetic datasets: demand points and results.**

assigned to a demand unit and in turn to a destination, this results in the blank unoccupied islands within the region. The process is explained in detail in *Gwalani, Tiwari & Mikler (2021)*. The value of $p$ is chosen to be approximately equal to $n/4$ to create large-scale problems.

We used three synthetic datasets to evaluate the EM-FI algorithm for proof of concept. Figure 5 shows the demand distribution in these datasets, the cost function and execution time results obtained for these datasets using the three heuristic approaches. It can be seen that the cost function values are reasonably good (within 2% of the best-known solution) for EM-FI even when a global improvement is not employed in the reduce step for the datasets with multiple clusters. Additionally, the EM-FI strategy is at least five-eight times faster than FI. The EM-FI:Reassignment does not perform as well for Dataset III because it contains only one large cluster and the decomposition breaks it into multiple parts even when they belong to the same GMM component which results in the poorer results. Using EM-FI:Improvement however, yields results that are within 0.5% of the best-known solution with a much lower execution time than the stochastic version of FI. In fact, statistical analyses using the Man-Whitney U test[3] showed that the cost function values obtained from EM-FI:Improvement are statistically significantly lower than the cost function values obtained using FI.

## Real datasets

We use response planning in preparation for bio-emergencies as an application of the p-median location model to evaluate the algorithm on real datasets. Public health emergencies can be catastrophic to the health and well-being of local communities. These emergencies require a timely and well-planned response from the local public health authorities to minimize casualties. The Centers for Disease Control and Prevention recommends setting up *ad-hoc* clinics or dispensing locations to distribute supplies and medical countermeasures (MCMs) to the affected population within a stipulated time frame (*Centers for Disease Control and Prevention, 2006*). We formulate this problem as the p-median problem where census block group centroids in the region represent population centers or demand points in the region, and the dispensing locations are the facilities that need to be located. The results are demonstrated for three regions in the United States with varying spatial distribution of the census block group centroids, 1) Dallas County, Texas, 2) Texas Department of State Health Services Region 6, Texas and 3) Los Angeles County, California. Figure 6 shows the demand point distributions, problem scales and results for these datasets. It can be seen that since there are more obvious distribution changes and the clusters are more spread out and distinct for Dallas and Los Angeles counties, EM-FI:Reassignment is able to produce reasonably good results (within 5% of the best-known solutions) with respect to the cost function values for these datasets. EM-FI:Improvement reduces the cost function values further for these datasets with relatively low execution time gain. The block group centroids in Region 6 are mostly centered around the city of Houston (Fig. 6B). The minimum BIC value was obtained for 18 components for this region. Figure 7A shows the decomposition of block groups in Region 6 into 18 parts. The components after the merge step is performed on the sparser

[3] The Mann-Whitney U rank test (*Mann & Whitney, 1947*) can be used to test if a randomly selected value from one population distribution is likely to be larger (or smaller) than a random value from another population distribution.

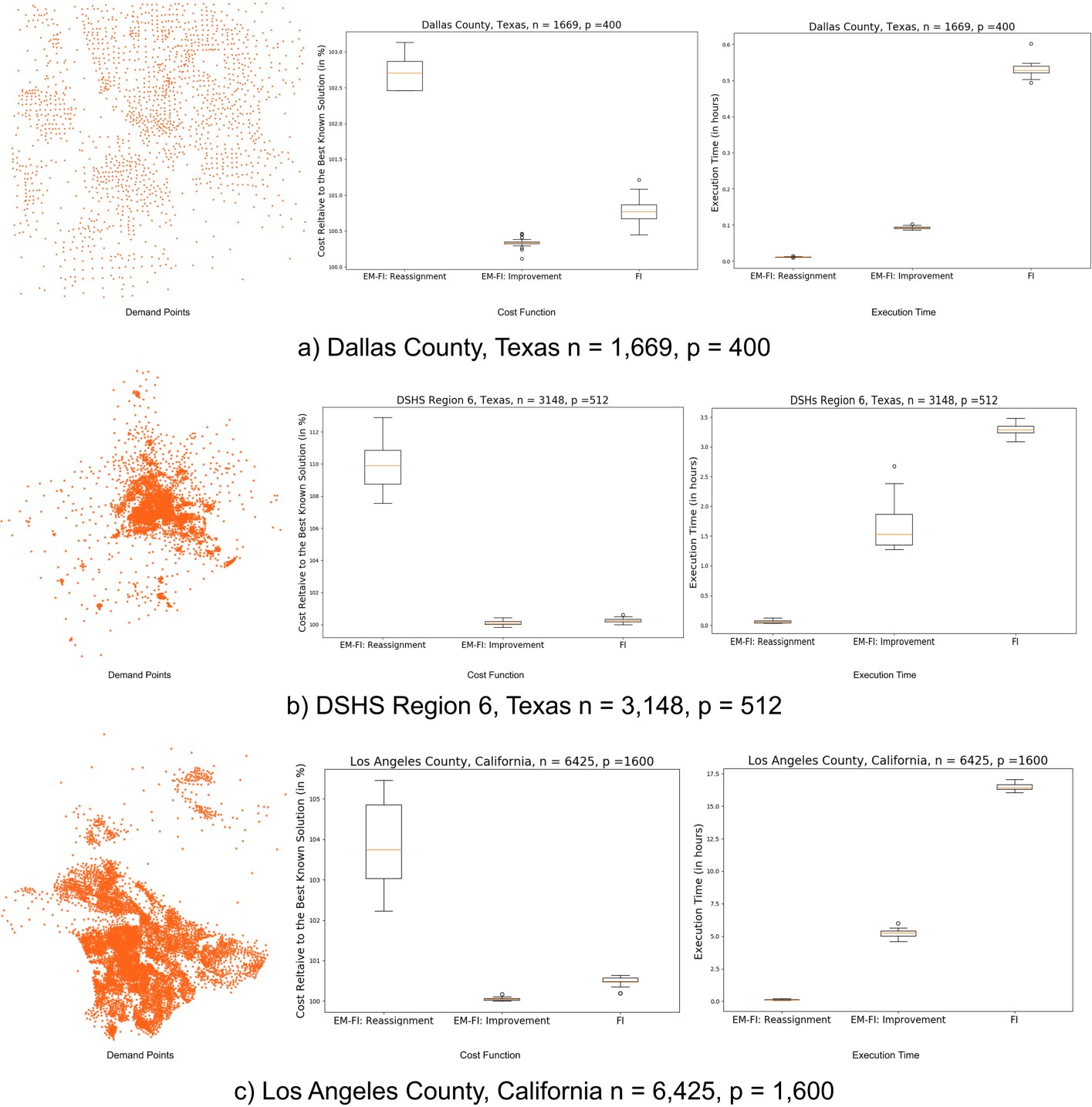

a) Dallas County, Texas n = 1,669, p = 400

b) DSHS Region 6, Texas n = 3,148, p = 512

c) Los Angeles County, California n = 6,425, p = 1,600

**Figure 6** (A–C) Real datasets: demand points and results. 

and homogeneous regions are shown in Fig. 7B. EM-FI:Reassignment produces poorer results because of the decomposition of the single large component in the center into subregions. Consequently, this causes the EM-FI:Improvement algorithm to take more

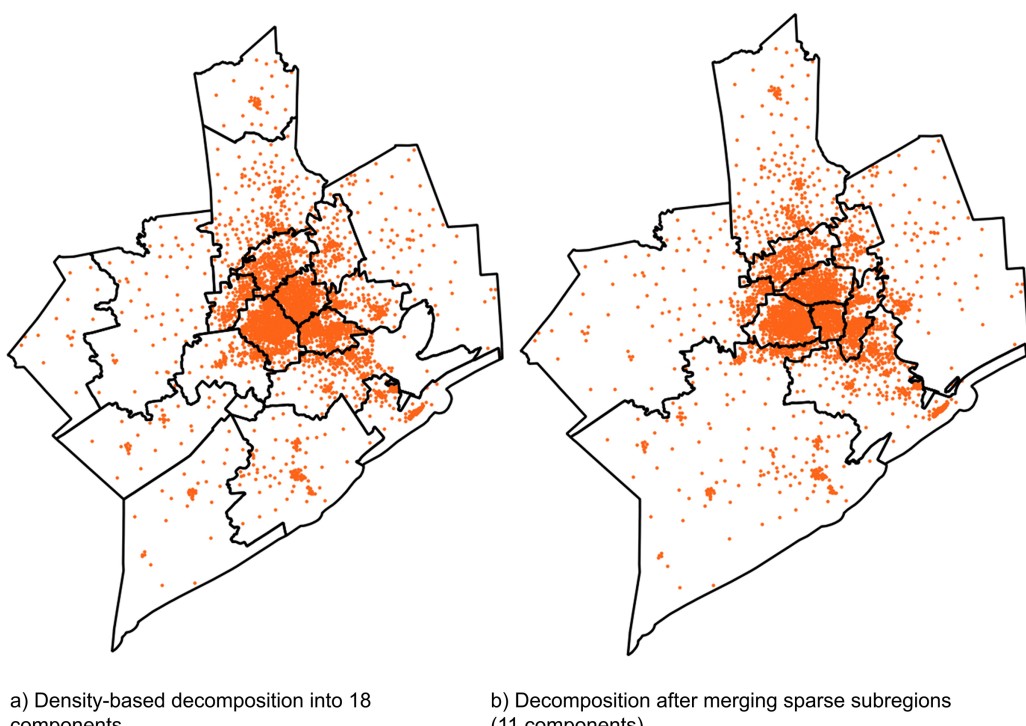

a) Density-based decomposition into 18 components

b) Decomposition after merging sparse subregions (11 components)

**Figure 7** (A and B) Density-based decomposition for Region 6, Texas.

**Table 2 Comparison of EM-FI:Reassignment with FI on an i7 dual quad core 32 GB machine.** The values in (Q1–Q3) represent the first and third quartile values of the relative cost values across multiple runs.

| Problem scale $(n, p)$ | EM-FI:Reassignment: Cost | EM-FI:Reassignment: Time | FI: Final cost | FI: Time |
|---|---|---|---|---|
| Dallas County, Tx (1,669, 400) | 102.46% (102.46–102.86)% | 0.011 (0.011–0.011) h | 100.44% (100.69–100.87)% | 0.529 (0.521–0.539) h |
| Region 6, Tx (3,148, 512) | 107.66% (108.86–110.97)% | 0.063 (0.042–0.078) h | 100.00% (100.08–100.16)% | 3.293 (3.235–3.347) h |
| Los Angeles County, CA (6,425, 1,600) | 102.22% (103.02–104.85)% | 0.124 (0.100–0.140) h | 100.19% (100.47– 100.57)% | 16.571 (16.42–16.71) h |

time as the initial solution in the reduce step is not that close to the optimal solutions (local or global). Despite this limitation, the EM-FI:Improvement algorithm outperforms FI with respect to both execution time and cost function values. Table 2 shows the comparison of results with respect to both execution time and cost function values between FI and EM-FI:Reassignment for these datasets. It can be seen that the execution time is almost two orders of magnitude less for the EM-FI:Reassignemnt when compared to FI even on a personal quad core machine. The execution time column in the table shows the average time across 30 runs and the first and third quartile values. The EM-FI:reassignment produces reasonably good solutions for clustered datasets even with respect to the cost function values (within 4% of the best solution). The cost column in the

table shows the minimum relative cost and the first and third quartiles values for relative costs.

## Challenges and distance-based decomposition

There are three drawbacks for using the EM-FI:Improvement algorithm for any general large-scale problem. (1) Over decomposition of a region with a single large cluster into subregions as a result of the EM-BIC analysis. The merge step may not combine the subregions belonging to the same dense cluster and lead to over-decomposition, as illustrated in Fig. 7. This over-decomposition yields to poorer results for this kind of distribution for EM-FI:Reassignment and increases the run time for EM-FI:Improvement. (2) Additionally, even for a multi-clustered, large problem, the final step in EM-FI improvement can prove to be a bottleneck in terms of execution time because of the size of the problem, even if the solution after EM-FI reassignment is close to the optimal solution. (3) For an entirely homogeneous distribution, the density based decomposition yields the original problem after the merge step, hence there are no time gains.

These three problems can be resolved by replacing the EM-FI improvement step with another decomposition step. The objective of this level of decomposition is to divide the current sources selected after the divide and conquer step (EM-FI reassignment) into $k$ parts using a distance based clustering algorithm like k-means. The idea behind this decomposition is that the destinations that are far away from a source are unlikely to be served by it given the distance based optimization function. The value of $k$ is determined by using the cluster quality criterion, Davies–Bouldin Index (*Davies & Bouldin, 1979*), on clusters created for $k = 2$ to $k = k_{max}$. The Davies–Bouldin Index selects the clustering solution that minimizes the ratio of intra-cluster dissimilarity (scatter) and inter-cluster dissimilarity. The index evaluates the ratio, $R_{ij}$, for all pairs of clusters, $C_i$ and $C_j$, $i \neq j$, in the solution. The intra-cluster scatter or the numerator of $R_{ij}$ is the sum of the average distance between all members in cluster $C_i$ to the centroid of $C_i$ and the average distance between all members in cluster $C_j$ to the centroid of $C_j$. While the inter-cluster dissimilarity between $C_i$ and $C_j$ is the distance between the two cluster centroids. For each cluster $C_i$, the maximum value of $R_{ij}$ is selected as its index, the Davies–Bouldin measure for the solution is the average index value across all clusters. A lower value of this index indicates better clustering. Each part is treated as an independent subproblem with the sources in that part as the initial solution to the subproblem for serving the union of destinations assigned to them. These subproblems are solved using FI concurrently, and since the initial solutions are already close to the optima these subproblems converge sooner. This step of distance based decomposition is repeated for $k = k - 1$ (until $k = 1$) by clustering the sources obtained from solving the subproblems created in the previous iteration. If the objective cost function value remains unchanged across two consecutive distance based decomposition iterations, then the process is terminated before $k = 1$. The final step in this process for $k = 1$ (if the algorithm does not terminate before that) is identical to the EM-FI Improvement step, however, the initial solution is now even closer to the optimum because of the previous decompositions, and hence it is much faster. We conducted experiments

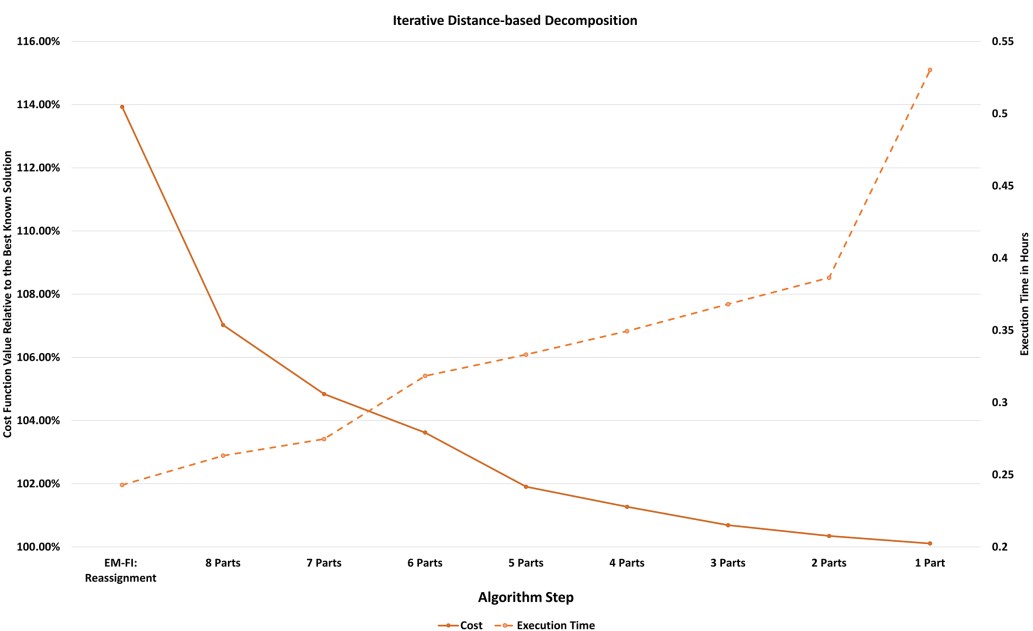

**Figure 8** Increase in execution time and decrease in cost function value for each iteration of distance-based decomposition after EM-FI:Reassignment for Region 6, Texas.

for the real dataset problems (10 runs for each dataset) on a Google compute machine (n1-highmem-16, 16 vCPUs, 104 GB RAM) employing the iterative distance based decomposition after EM-FI:Reassignment instead of the improvement step. We executed six MILP problems and 10 FI subproblems concurrently on this machine because of higher memory and CPU capabilities. Figure 8 shows the changes in execution time and relative cost function values with each iteration of distance based decomposition, after the reassignment step in a single execution for Region 6. Table 3 shows the cost and execution time results for these experiments and compares them with the results obtained from EM-FI:Improvement on the i7 dual quad core machine. The time gains from both the use of a more powerful machine and distance-based decomposition are shown in the table. Additionally, we solved a p-median problem to select 2,500 facilities in the entire state of Texas on the Google machine. There are 15,811 census block groups in the state of Texas. A problem of this scale would have taken days to execute if solved using the serial implementation of FI. A solution within 4% of the final solution was obtained in about 1.5 h for this problem and the final solution was obtained in less than 9 h.

If the density based decomposition yields the original problem because of the absence of clusters, the region can be divided into $k$ subproblems using the distance-based decomposition strategy described above. The first iteration solution can be improved by dividing the sources obtained from the first iteration into $k = k − 1$ parts iteratively. These decomposition strategies, however, may yield a single large subproblem that becomes the bottleneck to obtain a solution. This was observed in the case of the Texas dataset which

**Table 3 Comparison of cost function values and execution time for results on a dual quad core machine (PC) and a Google compute machine (GCP).** The values in (Q1–Q3) represent the first and third quartile values of the relative cost values across multiple runs.

| Problem scale $(n, p)$ | PC: Final cost | PC: Time | GCP: Final cost | GCP: Time |
|---|---|---|---|---|
| Dallas, Tx (1,669, 400) | 100.11% (100.32–100.34)% | 0.092 (0.090–0.094) h | 100.33% (100.34–100.37)% | 0.057 (0.053–0.059) h |
| Region 6, Tx (3,148, 512) | 100.00% (100.14–100.33)% | 1.64 (1.349–1.868) h | 100.00% (100.08–100.16)% | 0.403 (0.357–0.448) h |
| Los Angeles, CA (6,425, 1,600) | 100.00% (100.02–100.07)% | 5.22 (5.03–5.42) h | 100.01% (100.05–100.10)% | 1.87 (1.77–1.96) h |
| Texas, USA (15,811, 2,500)* | – | – | 100.00 % | 8.56 (8.56–8.56) h |

**Note:**
* Single run on Google Compute Engine.

caused the execution time to increase. The algorithm can be used recursively to overcome this problem.

Another challenge with the EM-FI decomposition algorithm is the unreliable nature of the BIC criterion. Higher numbers of clusters seemed to yield lower BIC values and lead to over-decomposition, especially in case of centered distributions. There are many cluster quality metrics and criteria that measure the intra-cluster similarity and the inter-cluster similarity in different ways. These metrics include cluster validation indexes such as Calinski and Harabasz score (*Caliński & Harabasz, 1974*), Dunn's Index (*Dunn, 1973*), Davies–Bouldin index (*Davies & Bouldin, 1979*), and silhouette coefficient (*Rousseeuw, 1987*), if the ground truth is not known. It was observed that these indexes yield vastly different results with varying distributions. BIC seemed to yield the best results for identifying density-based clusters and the Davies–Bouldin Index was selected to evaluate distance-based clusters. For a given region, denser regions can be identified by using DBSCAN without the dependence on the number of clusters, but this algorithm requires two additional parameters, minimum number of neighbors and the radius of neighborhood, which are not generalizable across problems as density is a relative concept. Additionally, the EM algorithm is faster than DBSCAN, hence more suitable for efficiency gains. The algorithm can be made even more efficient by employing parallel versions of expectation-maximization (*Lee, Leemaqz & McLachlan, 2016*) if necessary, although the expectation maximization step in the algorithm is much faster than the p-median steps, therefore the time gains are expected to be minimal.

# DISCUSSION

A novel distributed algorithm to solve large-scale p-median problems efficiently is presented in this article. The algorithm, EM-FI, utilizes the spatial configuration and the density distribution of demand points to decompose the region into subregions, and corresponding subproblems that can be solved concurrently without the loss in quality of the solution. The EM-FI:Reassignment heuristic, in which the union of the facilities selected in the subproblems is the solution to the original problem is at least five-eight times faster than FI and produces close to optimal solutions for datasets with distinct clusters. Performing FI on the EM-FI:Reassignment solution further improves the solution and yields better results than the FI heuristic both with respect to time and cost function

**Peer**J Computer Science

values. We solved p-median problems of much larger scales ($n > 3{,}000$, $p > 500$) than presented in existing research in less than 1 h on a personal computer. This research paves the way for solving nationwide/statewide location-allocation problems, such as the selection of Amazon inventory centers, or electric vehicle charging stations, across the country or state optimally in a reasonable amount of time.

The experiments for this study were performed without using a high performance cluster. The algorithm was distributed across cores/CPUs on the same machine. The run time can be reduced further by using a cluster of compute machines. The cluster can be set up in a way that enables the solution of a subproblem by MILP or parallel FI or EM-FI depending on the scale of the subproblem. This setup will ensure that no subproblem is a bottleneck. In the parallel version of FI, the swaps can be evaluated faster by comparing the exchange of an existing facility with all candidate facilities at the same time. Furthermore, better resources may enable solving all subproblems exactly using MILP solvers. Additionally, there is ongoing research on finding scalable methods to parallelize MIP solvers (*Ralphs et al., 2018*). Existing methods do not scale well enough for the original p-median problems, but these methods can be explored to solve the subproblems faster, if the resources are available. There is additional potential to compare the decomposition discussed in this study with the Bender's decomposition approach shared in *Duran-Mateluna, Ales & Elloumi (2023)* to create and solve p-median subproblems. Furthermore, there is scope to improve this study by extending the model to use real traffic data, accounting for the distance to raw materials (*Church, Drezner & Kalczynski, 2023*), and the exact locations of facilities (*Croci, Jabali & Malucelli, 2023*).

## APPENDIX

### Detecting heterogeneity in a region

The heterogeneity (or homogeneity) of the spatial distribution of points in a region is identified by overlaying a fine grid over the region. Let the number of demand points in the subregion be $n_{sub}$. The expected distribution for the number of cells in the grid with $i$ ($0 \leq i \leq n_{sub}$) demand points is compared with the obtained frequency distribution using the $\chi^2$-statistic. The probability that a demand point is located in any cell in a homogeneous distribution is $p_{cell} = 1/N$, $N$ is the total number of equal-sized cells in the grid. The probability that a cell will be occupied by $i$ demand points, $p(i)$ is equal to

$p(i) = \binom{n_{sub}}{i} p_{cell}^i (1 - p_{cell})^{n_{sub}-i}$. Therefore, the expected number of cells with $i$ demand points is $E(i) = Np(i)$. This distribution is compared with the actual number of cells with $i$ demand points, $F(i)$, in the subregion to check for deviation from a random distribution.

The $\chi^2$ statistic can then be calculated as $\chi^2 = \Sigma_{i=0}^{n_{sub}} \frac{(E(i)-F(i))^2}{E(i)}$. This $\chi^2$ probability distribution has $n_{sub}$ degrees of freedom. The homogeneity hypothesis is evaluated by calculating the probability that the $\chi^2$ value lies on the chi-square distribution function. We use the significance level of 5% for this analysis ($\alpha = 0.05$).

### Funding

This work was supported by the National Institutes of Health (Grant/AwardNumber: 2 R56 LM011647-03) and a contract with the Texas Department of State Health Services. The funders had no role in study design, data collection and analysis, decision to publish, or preparation of the manuscript.

### Grant Disclosures

The following grant information was disclosed by the authors:
National Institutes of Health: 2 R56 LM011647-03.
Texas Department of State Health Services.

### Competing Interests

Armin R Mikler is an Academic Editor for PeerJ.

### Author Contributions

- Harsha Gwalani conceived and designed the experiments, performed the experiments, analyzed the data, performed the computation work, prepared figures and/or tables, authored or reviewed drafts of the article, and approved the final draft.
- Joseph Helsing performed the experiments, analyzed the data, performed the computation work, prepared figures and/or tables, and approved the final draft.
- Sultanah M. Alshammari analyzed the data, prepared figures and/or tables, authored or reviewed drafts of the article, and approved the final draft.
- Chetan Tiwari conceived and designed the experiments, authored or reviewed drafts of the article, and approved the final draft.
- Armin R Mikler conceived and designed the experiments, authored or reviewed drafts of the article, and approved the final draft.

### Data Availability

The code and results are available at GitHub and Zenodo:
- https://github.com/harsha2412/EM-FI
- harsha2412. (2024). harsha2412/EM-FI:v1 (Version v1). Zenodo. https://doi.org/10.5281/zenodo.13371696.

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
