# Peer review of "A distributed algorithm for solving large-scale p-median problems using expectation maximization"

_PeerJ Computer Science, doi:10.7717/peerj-cs.2446_

## Round 0.1 · original submission · Major Revisions

Reviewers agree on the relevance of your contribution, pointing out some aspects that needs to be addressed for improving it. Please address them, in particular the inclusion of an heuristic methods in the experimental results and improve the quality of images.

Reviewer 1 ·

Basic reporting

This study presents an algorithm for solving large-scale p-median problems. The algorithm identifies the existing spatial clusters of the destination locations using EM. The key point of this manuscript is to decompose large-scale problems into many sub-problems through partitioning, then solve the sub-problems using integer programming or FI concurrently. The algorithm provides a feasible approach for solving large-scale p-median problems, but there are still some issues.
1. In line 64, the text states, "A demand point belonging to a cluster is not served by a facility located in another cluster." This is an intriguing observation. If it is a previously expressed viewpoint, it would be beneficial to indicate the source.
2. 2. In Figure 1, there are some unknown blocks. If it is deemed appropriate, please elucidate their significance.
3. All the images in the manuscript are of insufficient quality when enlarged. It is therefore requested that a clearer image be provided.
4. Some classic algorithms are mentioned in the literature review. In recent years, some new methods have also been applied to the p-median problem.

Experimental design

5. As the author noted, numerous algorithms exist for solving the p-median problem, including LP solvers and heuristic algorithms. Incorporating a heuristic method, such as a genetic algorithm, would enhance the credibility of the experiment.

Validity of the findings

6. The data and code were not obtained.

Additional comments

In summary, this work presents an interesting approach to the large-scale p-median problem.

Reviewer 2 ·

Basic reporting

• The manuscript is clearly written in professional, unambigous language.
• Literatures are well referenced and relevant
• Figures are relevant, high quality, well labelled & described.
• Structure conforms to PeerJ standards, discipline norm.
• Raw data supplied

Experimental design

• Original primary research within the Scope of the journal.
• Research question well defined, relevant& meaningful. It is stated how the research fills an identified knowledge gap.
• Methods described with sufficient detail & information to replicate

Validity of the findings

• All underlying data have been provided.
• Conclusions are well stated, linked to original research question.

Additional comments

The content of the article is about dividing the p-median problem into smaller subproblems. First, n points are divided into disjoint subsets based on density (Expectation Maximization, EM) and sparse regions are grouped with nearby dense regions. The p facilities are also distributed among these subsets, proportional to the points in each subset. Then, the p_i median subproblems are solved using Mixed Integer Linear Programming (MILP). If solving with MILP takes too much time, an approximate solution using the Fast Interchange (FI) algorithm is used. After solving the small problems, the n points are reassigned to the p facility locations that were just found, with each point assigned to the nearest facility (EM-FI: Reassignment). The approximate solution found can be further improved by running the FI algorithm starting from the just-found solution (EM-FI: Improvement). The author also points out three drawbacks of this method: (1) if the dense regions are very close to each other, the solution found will not be good enough, (2) the Improvement step can take a lot of time, and (3) if the points are evenly distributed, the proposed algorithm will not be faster than existing algorithms. To solve these three issues, the author proposed dividing the subsets based on distance right after the Reassign step.
Minor comments:
a. Your primary concern pertains to the benchmarking datasets. As indicated in the manuscript (lines 41 and 456), your focus lies on problems involving n > 3000 and p > 500. However, it’s worth noting that two out of three problems occur in synthetic datasets, while one out of four problems occurs in real datasets with n < 3000.
b. In the benchmarking datasets, you consistently select p to be equal to n/4. However, if your goal is to generate data with n > 3000 and p > 500, choosing p = n/6 would be a more suitable option. It’s important to note that in real-world scenarios, the value of p is typically much smaller than n/4.
c. In line 47, you express interest in problems with n > 5000. However, this contradicts what you state in lines 41 and 456.

Annotated reviews are not available for download in order to protect the identity of reviewers who chose to remain anonymous.

---

## Round 0.2 · accepted · Accept

The authors addressed the comments, as discussed in the rebuttal letter and verified by one reviewer. The manuscript now is ready for publication.

Reviewer 2 ·

Basic reporting

The writing is clear.

Experimental design

Experiemental design is acceptable

Validity of the findings

Findings are novel and interesting.

Additional comments

I recomend: Accept